# Master Regulator Analysis of the SARS-CoV-2/Human Interactome

**DOI:** 10.3390/jcm9040982

**Published:** 2020-04-01

**Authors:** Pietro H. Guzzi, Daniele Mercatelli, Carmine Ceraolo, Federico M. Giorgi

**Affiliations:** 1Department of Surgical and Medical Science, University of Catanzaro, 88100 Catanzaro, Italy; hguzzi@unicz.it; 2Department of Pharmacy and Biotechnology, University of Bologna, 40126 Bologna, Italy; daniele.mercatelli2@unibo.it (D.M.); carmine.ceraolo@studio.unibo.it (C.C.)

**Keywords:** coronavirus, bioinformatics, gene network analysis, COVID-19, SARS-CoV-2

## Abstract

The recent epidemic outbreak of a novel human coronavirus called SARS-CoV-2 causing the respiratory tract disease COVID-19 has reached worldwide resonance and a global effort is being undertaken to characterize the molecular features and evolutionary origins of this virus. In this paper, we set out to shed light on the SARS-CoV-2/host receptor recognition, a crucial factor for successful virus infection. Based on the current knowledge of the interactome between SARS-CoV-2 and host cell proteins, we performed Master Regulator Analysis to detect which parts of the human interactome are most affected by the infection. We detected, amongst others, affected apoptotic and mitochondrial mechanisms, and a downregulation of the ACE2 protein receptor, notions that can be used to develop specific therapies against this new virus.

## 1. Introduction

Beta-coronaviruses are a subgroup of the coronavirus family, large enveloped positive-stranded RNA viruses able to infect a wide variety of mammals and avian species, causing mainly respiratory or enteric diseases [1]. They present surface spikes formed by (S) glycoproteins, which mediate binding to susceptible host cell receptors to initiate the infection [2]. To date, seven beta-coronaviruses are known to cause human disease. The prevalent strains (HCoV 229E, HKU1, NL63 and OC43) typically cause mild infections of the upper respiratory tract in humans [3], while three strains were found to pose a severe health threat, leading in some cases to a fatal outcome: SARS-CoV, MERS-CoV and the newly identified SARS-CoV-2 [4]. SARS-CoV was recognized as the causal agent of the severe acute respiratory syndrome outbreak occurring in China in 2002 and 2003 [5], while MERS-CoV was responsible for the Middle East respiratory syndrome occurring in 2012 [6]. In both cases, the infected patients manifested severe acute pneumonia, but while SARS-CoV infected mainly the lower respiratory tract, MERS-CoV caused more pronounced gastrointestinal symptoms, often associated with kidney failure [7]. This can be possibly explained by a difference in recognition of host surface receptors. While the primary human receptor for SARS S glycoprotein is the angiotensin-converting enzyme 2 (ACE2), which is widely expressed on the surface of the cells in the lower respiratory tract, MERS-CoV preferentially binds to surface enzyme dipeptidyl peptidase 4 (DPP4, also known as CD26), which is present both in the lower respiratory tract as well as the kidneys and the gastrointestinal tract [8,9,10]. This distribution may also explain the observed difference in transmission of the two viruses [7]. The third highly pathogenic strain of coronavirus, SARS-CoV-2, appeared in late December 2019 in Wuhan, Hubei Province, China [11]. It is a highly aggressive new coronavirus that has infected more than 540,000 people (at 27 March 2020), with an overall case-fatality rate of 2.3% [12]. In less than two months, the SARS-CoV-2 outbreak spread from the Hubei Province to 26 other countries, reaching also the US and Europe, with Italy being the European country with the highest number of patients tested positive for the new coronavirus [13] with 80,589 affected people and 8215 deaths by 27 March 2020. Given the emergency, the characterization of the biology of SARS-CoV-2 and features of virus-host interactions demand that we identify the candidate molecules to inhibit viral functions. Sequence similarity of new coronavirus with SARS and MERS was previously reported and close similarities with wild bat coronaviruses has been reported [14]. New coronavirus spike S glycoprotein sequence shares a sequence identity above 72% with human SARS, showing a unique furin-like cleavage site, which is a feature absent in the other SARS-like CoVs [15]. Biophysical and structural evidence that new SARS-CoV-2 may bind the ACE2 receptor with higher affinity than SARS-CoV has been recently reported [9]. The wide distribution and similarity of this receptor in the animal kingdom may account for cross-species transmission [16], while the pattern of expression of ACE2 in human respiratory epithelia and oral mucosa may explain the fast human-human transmission [17].

Despite the crucial need for a clinical strategy to counteract this epidemiological outbreak, no specific drug has been identified yet. This will require the elucidation of genetic features, molecular constituents and mechanisms of infection, which ultimately reside in the mechanisms regulating virus-host interaction and host responses. However, there is a lack of molecular data describing the COVID-19, and those available are related to the genomic and protein sequence of this virus [14]; there is currently no high-throughput dataset describing the mechanism of infection, e.g., transcriptomics of infected lung cells.

In this context, the use of bioinformatics approaches may help to fill the gap of missing data and to improve the process of knowledge discovery by elucidating molecular mechanisms behind virus replication, the process of viral attachment to the host cells and the effect on the host molecular pathways [18]. The final understanding of the molecular action of the virus may improve or accelerate the development of anti-viral therapeutic approaches using information and data available for other coronaviruses [19].

Here, we applied a network approach to investigate cellular responses to the interaction of human cells with SARS and MERS viruses to get more insights into the molecular mechanisms regulating susceptibility to infection and transmission. The analysis of cell-context specific repertoires of molecular interactions, or interactomes, is becoming increasingly useful to dissect molecular pathways and biological processes, since they offer the possibility to get a comprehensive overview of cellular regulatory programs [20]. Integration of interactome maps to gene-regulatory networks (i.e., topological maps representing relationships between transcription factors and their target genes) can better provide insights into critical cellular elements that regulate a certain phenotypic state or response [21]. It has been observed that a number of large-scale transcriptional cascades behind complex cellular processes can be explained by the action of a relatively small number of transcription factors, or Master Regulators [22,23,24]. The identification of the most crucial factors governing a determined phenotypic state is known as Master Regulator Analysis (MRA) [25]. In this study, we set out to apply an integrated network approach combining protein interactions and transcriptional networks to identify Master Regulators of coronavirus infections in human airway cells, in the hope of shedding more light on the virus/host interaction mechanism and also on the evolutionary origin of SARS-CoV-2.

## 2. Materials and Methods

### 2.1. Interactome Model for SARS-CoV-2/Human Cell Interaction

We obtained a recently generated interactome of all the physical interactions between SARS-CoV-2 and human proteins based on the work by Korkin and colleagues [26]. In brief, the interactome contains both virus-virus and virus-host protein-protein interactions based on structural interaction modeling and established experimental knowledge. The resulting interactome consists of 125 proteins (31 viral proteins and 94 human host proteins) and 200 unique interactions. The interactome is available at http://korkinlab.org/wuhan.

### 2.2. Coexpression-Based Lung Network

Coexpression-based networks [27] were generated using the corto algorithm with default parameters, freely available on the CRAN repository of R packages [28], using the SARS-CoV-2/human interactome proteins derived from [26] and the largest human lung RNA-Seq dataset available from the GTEX consortium, which was generated from 427 patients gene expression profiles [29]. In brief, corto calculates a coexpression network for each protein and then removes indirect interactions using Data Processing Inequality [30], which has been shown to provide a more robust readout of single protein abundance alone [31].

### 2.3. Datasets of MERS and SARS Infection

We obtained two datasets describing transcriptome-wide effects of coronavirus infection of human cultured 2B4 bronchial epithelial cells. The first dataset (available as ArrayExpress [32] dataset E-GEOD-17400) measured SARS-CoV infection using the microarray platform Affymetrix Human Genome U133 Plus 2.0 [33], producing nine infected samples and nine mock samples. The dataset was normalized using RMA [34], and probeset mapping was performed using the most updated annotation from CustomCDF v24.0.0 [35]. The second dataset (available as ArrayExpress dataset E-GEOD-56677) measured gene expression upon MERS-CoV infection using Agilent-039494 SurePrint G3 Human GE v2 8 × 60K microarrays [36] and contained 18 infected samples and 15 mock samples.

### 2.4. Master Regulator Analysis

Master Regulator Analyses were performed by comparing infected and mock samples in both MERS and SARS datasets separately with the *corto* algorithm [28] using default parameters and the coexpression network derived from human lung samples. In brief, a gene-by-gene signature of viral-induced differential expression is generated, and a combined value for each coexpression network is generated by weighing every gene’s likelihood in the network, providing a final Normalized Enrichment Score for each human/SARS-nCoV-2 interactome member, which is positive if the network is upregulated by the infection, and negative if it is downregulated, as in [37].

### 2.5. BLAST

Pairwise protein identity and coverage were calculated using the BLAST protein v2.6.0 [20] with BLOSUM62 matrix and default parameters. Nucleotide sequence identity and coverage were calculated using BLAST nucleotide v2.6.0 [20].

### 2.6. Phylogenetic Analysis of Viral Genomes and ACE Orthologs

A total of 201 genome sequences were obtained from GISAID and GenBank. Specifically, we obtained 8 pangolin coronavirus sequences, 6 bat coronavirus sequences, 6 SARS coronavirus sequences, 3 MERS coronavirus sequences and all 177 SARS-nCoV-2 genome sequences available on 25 February 2020. A full description of all GISAID sequences is available in Appendix A.

To obtain the pangolin virus sequence PRJNA573298, we performed a deeper data mining analysis. FASTQ sequences were retrieved from the Sequence Read Archive (SRA) using SRA-toolkit v2.9.6-1 [38] and using project PRJNA573298 run ids SRR10168377 and SRR10168378, which provided a DNA readout of the pangolin viral metagenome [39]. Reads were mapped over the reference 2019-nCoV genome NC_045512.2 using Hisat2 v2.1.0 [40] with default parameters (with option-no-spliced-alignment). The environment programs were samtools v1.9 and bedtools v2.26.0. Genome assembly of the virus genome-mapping reads was performed using Abyss v2.1.5 with default parameters [41].

We collected the following ACE2 orthologous sequences from the NCBI Protein database, using representative organisms for major vertebrate groups. For mammals: NP_001358344.1 (*Homo sapiens*), XP_016798468.1 (*Pan troglodytes*), NP_001129168.1 (*Macaca mulatta*), NP_081562.2 (*Mus musculus*), XP_017505752.1 (*Manis javanica*), AGZ48803.1 (*Rhinolophus sinicus*), NP_001012006.1 (*Rattus norvegicus*), XP_005228485.1 (*Bos taurus*), NP_001116542.1 (*Sus scrofa*), XP_007500935.1 (*Monodelphis domestica*) and XP_001515597.2 (*Ornythorincus anatinus*). For birds: XP_416822.2 (*Gallus gallus*). For reptiles: XP_025066628.1 (*Alligator sinensis*), XP_032082934.1 (*Thamnophis elegans*) and XP_007070561.1 (*Chelonia mydas*). For amphibians: XP_002938293.2 (*Xenopus tropicalis*). For fish: XP_005169416.1 (*Danio rerio*), XP_005997915.2 (*Latimeria chalumnae*) and XP_007889845.1 (*Callorhinchus milii*).

Multiple Sequence Alignment (MSA) was performed using MUSCLE v3.8.31 [42] and is available in FASTA format as Appendix A. MSA visualization was generated via Jalview v2.11.0 [43]. Phylogenetic model generation and tree visualization was done using MEGAX v10.1.7 [44], using the Maximum Likelihood method and Tamura-Nei model [45]. The tree structure was validated by running the analysis on 100 bootstrapped input datasets [46]. In the viral genome trees (Appendix A), but not in the protein tree, branching points with bootstrap validation below 30% were collapsed. The tree in Appendix A collapsed the SARS-nCoV-2 subtree, but a full tree is available at Appendix A.

### 2.7. 3D Structural Analysis

The protein structure representation has been produced with Chimera v1.14 [47] based on Protein Data Bank structure id 1R42 [48].

## 3. Results

### 3.1. Human/COVID-19 Interactome Response to Coronavirus Infection

We obtained the most up-to-date specific SARS-CoV-2/human interactome from [26]. This network, which we represent in Figure 1, highlights the complex interaction between the 31 viral proteins (coded by a genome of roughly 30kb) and 94 human proteins, ranging from surface enzymes (ACE2) to RNA-processing proteins (DDX5) to mitochondrial constituents (BCL2) [49]. In order to understand how this interactome is functionally affected by the viral infection, we obtained transcriptome wide microarray data depicting gene expression changes upon MERS-CoV [36] and SARS-CoV [33] on human bronchial-derived 2B4 cells. We, therefore, performed Master Regulator Analysis (MRA) [22] using a context-specific network model derived from the largest available human lung RNA-Seq dataset from GTEX [29]. The analysis allowed us to have a robust readout not only of the changes of the transcript of each protein-encoding human gene in the interactome, but also of its proximal functional network [31]. Since transcriptomics data of SARS-CoV-2 infection are not yet available, we reasoned that combining the SARS-CoV-2 specific interactome with transcriptome-wide datasets from two viruses belonging to the same family would provide a robust readout of the human cell response to this new virus.

The combined results of both MERS-CoV and SARS-CoV infection are depicted in Figure 1 using Stouffer integration of the Normalized Enrichment Scores (NES) of both analyses. Individual MRAs of beta-coronavirus infections are available in Figure 2 for MERS-CoV analysis (Figure 2A) and SARS-CoV analysis (Figure 2B) for the eight protein networks with the highest integrated NES. Figure 3 shows the compared MRAs of MERS-CoV and SARS-CoV, and we decided to focus on the eight proteins that share the highest response similarity in both experiments, as they are more likely to be conserved host responses to beta-coronavirus infection. Full results for all human interactome proteins are available in Appendix A for SARS-CoV, MERS-CoV and both integrated by Stouffer integration, with the corresponding *p*-value.

Our MRA results show a strong, shared upregulation of MCL1, a positive regulator of apoptosis [48,50]. Apoptosis is a complex molecular cascade usually activated to eliminate unwanted cells in a living organism, for example during development, and it can be self-initiated or induced by immune system cells [51]. Also, during virus attacks, induced cell death is a known mechanism to get rid of infected cells: specifically, the activation of MCL1 in coronavirus infection has been observed as an early host cell mechanism of antiviral defense [52]. It is, therefore, likely that the host cell is able to sense the virus, perhaps even by the direct interaction between MCL1 and the viral ORF7a (Figure 1), and able to start the apoptotic cascade and shut down both itself and the replicating viruses. An increase in mitochondria-mediated apoptotic response upon SARS-CoV infection has been reported before, although its prevention did not seem to affect viral replication kinetics [53]. However, apoptosis has been reported to help the spreading of the virus by increasing replication rates and virion release from the dying cell in avian coronaviruses [54].

We also report several concordantly human proteins down-regulated by beta-coronavirus infections. One of these is EEF1A1 (Elongation factor 1-alpha 1), which is involved in tRNA delivery to the ribosome, and is known to be activated upregulated upon inflammation [55]. This protein has been shown to physically interact with proteins of several viral species, specifically hepatitis delta [56] and avian reoviruses [57]; in the latter, the knock-down of EEF1A1 has been shown to significantly impair the virus infection cycle. Therefore, the downregulation of EEF1A1 could be the result of an innate cell strategy to deprive SARS-CoV-2 of a key support for RNA replication.

Similar to EEF1A1, the NDUFA10 protein network is downregulated by viral infection. NDUFA10 is a member of the mitochondrial complex I [58]. This protein is known to be shut-down upon viral infection of lung A549 cells by human respiratory syncytial virus (HRSV), a paramyxovirus, also RNA-based [59]. The down-regulation of mitochondrial components in both coronaviruses and paramyxoviruses shows the conservation of RNA viral strategies to attack the host cell, which passes through the disruption of mitochondria.

Another down-regulated network is centered around the T cell energy-related E3 ubiquitin ligase RNF128, also called GRAIL [60], which has been reported to provide positive reinforcement of antiviral immune response to RNA viruses [61]. In this case, the inactivation of RNF128, which gets physically bound by two viral nonstructural proteins, NSP7 and NSP13, could be a coronavirus resistance mechanism against the cell innate defense.

MRA shows up-regulation (significant in MERS infection, not significant in SARS) of DDX5, DEAD-box polypeptide 5, a prototypical member of the RNA helicase family. DDX5 plays multifunctional roles and is involved in all aspects of RNA metabolism [62]. Since SARS-CoV-2, like all coronaviruses, is a single strand RNA virus, it uses both its own RNA-dependent RNA polymerase and host proteins to promote the replication of its genetic material. The up-regulation of DDX5 could be another mechanism of viral-triggered cell activation, especially since this protein has been shown to promote the infection capability of encephalitis flaviviruses [63], which are also RNA viruses.

Beyond the top eight protein networks influenced by SARS/MERS infection, our analysis shows that Transmembrane Serine Protease 2 (TMPRSS2), a protein recently proven fundamental for SARS-CoV-2 entry in the cell [64], interacts with the viral Spike protein S (Figure 1), and it is downregulated by beta-coronavirus infection (Figure 3), although at borderline significance and not amongst the top eight (*p* = 0.05, Appendix A).

It is important to note that all protein activities are calculated based on physiological coexpression network models, but they are based on largely non-overlapping coexpressed genes, thanks to the DPI step in the corto algorithm. This allows us to identify complementary signatures and co-responding networks based on independent gene measurements.

### 3.2. ACE2: Implications for Coronavirus Origin

Perhaps the most well-known human protein in relation with beta-coronavirus infection is ACE2 (Angiotensin Converting Enzyme 2), a zinc metalloprotease able to hydrolyze angiotensin I, angiotensin II, apelin-13, dynorphin A 1-13 and 7 additional small peptides [65]. ACE2 is exploited by SARS-CoV as its cellular entrance, which is based on the initial interaction between the viral S (spike) protein and ACE2 [66]. Several recent studies have also shown that SARS-CoV-2 entry in human cells depends on the interaction between the S protein and ACE2 [64,67]. Its down-regulation upon SARS-CoV and MERS-CoV infection could intuitively be an innate cell defense mechanism to remove this extracellular marker to reduce the capability of the virus to enter the host cell, while at the same time reducing the RNA metabolic machinery through EEF1A1 down-regulation. ACE2 is also targeted by the neuraminidase protein of the influenza A virus [68]. However, the down-regulation of ACE2 has also been shown to be beneficial to viral infection, as it can trigger lung injury and faster cell-to-cell virus spreading [69]. In influenza A (cause by orthomyxoviruses, also RNA-based), high levels of ACE2 have in fact been correlated to increased resistance to the infection [68]. So, like in apoptosis induction, the down-regulation of ACE2 could be a SARS-CoV-2-induced mechanism from which the virus gains in capability to spread faster by damaging the host lung tissue.

Our analysis so far has shown effects on human proteins interacting with SARS-CoV-2 based on SARS-CoV and MERS-CoV signatures. We conjectured that proteins similarly affected by both infections (like ACE2) could have a similar role in all beta-coronavirus infections, including SARS-CoV-2. Our analysis confirms previous reports [14] that SARS-CoV and MERS-CoV are indeed highly related to SARS-CoV-2, with genome sequence identity of ~80%. However, these two human viruses are likely not the most direct ancestors of SARS-CoV-2, as highlighted by our phylogenetic analysis based on several beta-coronaviruses ([14] and Appendix A). The evolutionary history model shows that SARS and MERS viruses are more distantly related to SARS-CoV-2 than wild beta-coronaviruses infecting bats and pangolins, in concordance with current evolutionary models [70], suggesting an almost certain zoonotic origin of this virus [71]. Our analysis shows that at least one bat beta-coronavirus sequence derived from the GISAID database (EPI_ISL_402131) appears to be the closest to the group of human SARS-CoV-2 genomes. However, this is based on a low bootstrap tree branching confidence (37%), and we observed a large quantity of pangolin beta-coronaviruses [72], which are likely to be equidistant from the human causes of COVID-19.

Based on our confirmation that ACE2 is an extremely important host protein for viral infection, we set out to compare ACE2 sequences in both bats and pangolins, to detect the most similar sequence to human ACE2. We obtained representative sequences from human ACE2 (NP_068576.1), pangolin ACE2 (*Manis javanica* XP_017505752.1) and bat ACE2 (AGZ48803.1, from the species *Rhinolophus sinicus*, being the closest species with a full protein sequence to *Rhinolophus affinis*, where the closest bat_CoV was retrieved). These sequences appear to be conserved in these species, having the exact same aminoacid length (805aa). The sequence identity between the human and pangolin ACE2 is 84.76%, while the identity between human and bat ACE2 is lower: 80.60%. Bootstrapped maximum-likelihood phylogenetic analysis of the vertebrate ACE2 family also shows that the pangolin ACE2 protein is closer to the human ortholog than the bat sequence (Figure 4A). In Figure 4B, we provide a visualization of the human ACE2 structure, inferred via X-ray crystallography at 2.20 Å, which is one of the highest resolutions for this protein [73], and highlight all the residuals identical in human and pangolins, but different between humans and bats. Finally, the Multiple Sequence Alignment (MSA) of the proteins from the three mammals reports the specific ACE2 amino acids that differ between the species (Appendix A). The higher similarity between human and pangolin ACE2 proteins may support the hypothesis that the pangolin could either be the original host of SARS-CoV-2, or an intermediate host for the transmission of the virus from a wild reservoir (e.g., bats→pangolins→humans).

## 4. Discussion

Our analysis integrates genome-wide and transcriptome-wide complementary network information on SARS-CoV-2/human interaction to provide a map of host proteins affected by the viral infection. Given the importance of finding new targets to effectively treat this new virus, understanding the molecular effects of this virus on human proteins might be pivotal in prioritizing pharmacological strategies. This approach can be extremely helpful given the observed clinical inefficacy of common antiviral drugs [74,75] or anti-inflammatory drugs [76].

One of the SARS-CoV-2-interacting protein networks most down-regulated by beta-coronaviruses in our analysis was ACE2, an experimentally validated molecular target used by SARS-CoV-2 for entry in the cell [64]. Administration of recombinant ACE2 to compensate for this protein loss has been shown to be effective in treating acute respiratory distress syndrome, which is often caused by severe pulmonary infections [77]. Recombinant ACE2 has also been proposed as a potential cure for SARS-CoV-2’s most severe symptoms [78], and in vitro studies have shown that soluble forms of ACE2 are beneficial to SARS patients, likely because they act as competitive binders of SARS-CoV Spike proteins, preventing binding to the host cell ACE2 [79]. Unpublished experimental results report recombinant ACE2 to also be a molecular neutralizer of SARS-CoV-2 [80].

Another membrane protein pivotal to viral entry in the cell is TMPRSS2, shown in our interactome and down-regulated by viral infection. The TMPRSS2 node is a potentially interesting region of the interactome, as it has been experimentally demonstrated that inhibition of TMPRSS2 with a protease inhibitor (camostat mesylate) inhibits SARS-CoV-2 replication [64], making this protein an extremely promising target for antiviral therapy.

Another point of clinical intervention could be to down-regulate the apoptotic mechanisms induced by the virus, e.g., by our reported MCL1 activation upon both SARS-CoV and MERS-CoV infections. As we reported, this response is in agreement with previous findings discussing the involvement of MCL1 upregulation in viral replication across many RNA-based viruses [52], possibly resulting from protein synthesis impairment eliciting ER-stress-induced apoptosis via unfolded protein response (UPR) [81]. Since beta-coronavirus-induced UPR has been shown to modulate innate immunity [82], targeting aberrant monocytes/macrophages activation may be another possible intervention strategy to reduce the severity of coronavirus-induced symptomatology.

We also highlighted how mitochondrial proteins interacting with SARS-CoV-2 appear to be downregulated in beta-coronavirus infections as a general viral mechanism to control the host cell metabolism and ultimately steering apoptosis [83]. The goal of preventing mitochondria-induced apoptosis is shared by clinical strategies to slow neurodegenerative pathologies, mainly Alzheimer’s disease, where mitochondrial dysfunctions are directly linked to neuronal death [84].

Our study highlights several other proteins that could be directly repressed or enhanced by specific drugs. For example, a mild virus-activated protein in our network is the serine/threonine kinase MARK2 (Figure 1), which is one of the targets of the drug Fostamatinib in the treatment of Rheumatoid Arthritis and Immune Thrombocytopenic Purpura [85]. Apart from drug repurposing [86], the recent advances in drug discovery by virtual screening [87] give scientists the chance to quickly develop novel molecules to specifically target not only viral proteins, but also the human proteins on whose interaction SARS-CoV-2 depends for infection and replication [88].

## Figures and Tables

**Figure 1 jcm-09-00982-f001:**
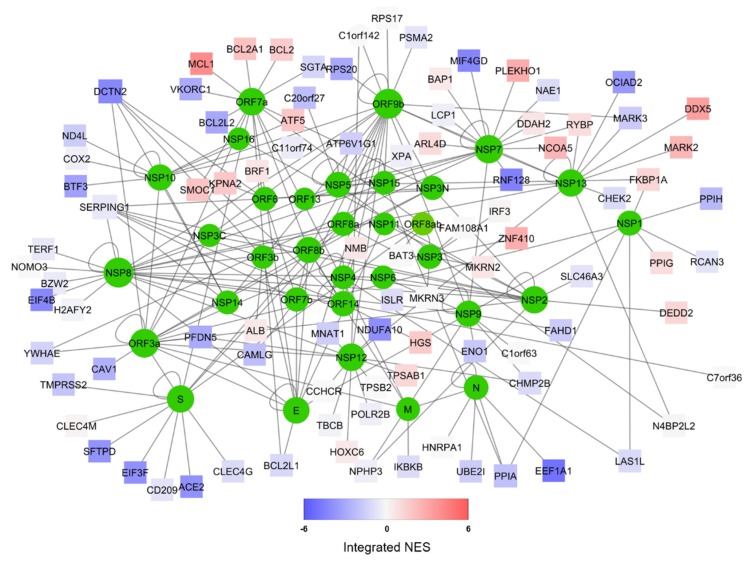
Representation of the predicted SARS-CoV-2/Human interactome [26] (available for download at http://korkinlab.org/wuhanDataset), containing 200 unique interactions among 125 proteins (nodes). SARS-CoV-2 proteins are depicted as green circles, while human proteins are represented as squares. The color of human protein nodes reflects the integrated effect of MERS and SARS infections on the node network (see Appendix A) as a Normalized Enrichment Score (NES). Network visualization was performed via Cytoscape [49].

**Figure 2 jcm-09-00982-f002:**
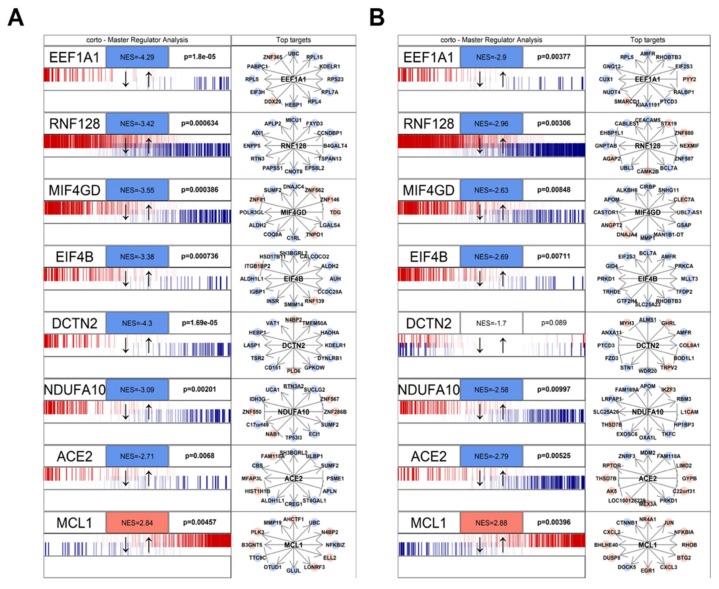
Master Regulator Analysis of the 8 human proteins in the human/SARS-CoV-2 interactome and most concordantly affected by beta-coronavirus infection. The visualization was obtained through the mraplot function of the R CRAN package *corto*. In brief, for each analyzed network, the centroid is indicated by its gene symbol (e.g., EEF1A1, ACE2). The genes in each network (generated by the corto package from the GTEX healthy lung RNA-Seq dataset) are shown in a barcode-like diagram showing all transcriptome genes by means of their differential expression upon viral infection, from most downregulated (left) to most upregulated (right). Positively- (red) and negatively- (blue) correlated targets are overlayed on the differential expression signature as bars of a different color. Normalized Enrichment Score (NES) and *p*-value are also indicated. To the right, the 12 highest-likelihood network putative targets of each protein are shown, in red if upregulated, in blue if downregulated, with a pointed arrow if predicted to be activated by the centroid protein, and with a blunt arrow if predicted to be repressed. The figure shows two analyses based on the MERS infection signature (**A**) and on the SARS infection signature (**B**). Full results are available as Appendix A.

**Figure 3 jcm-09-00982-f003:**
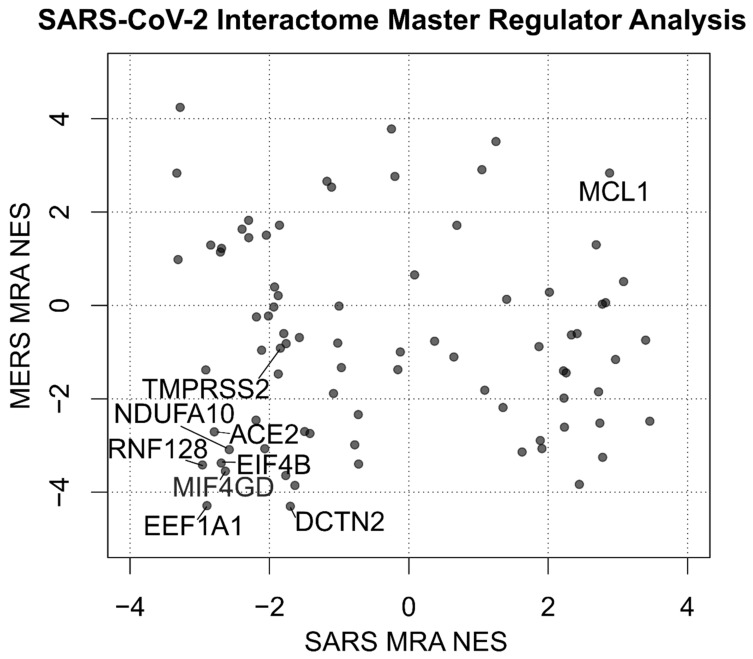
Scatterplot depicting the Normalized Enrichment Scores (NES) of the Master Regulator Analysis of human proteins interacting with SARS-CoV-2. Two analyses are compared using the signatures of MERS and SARS infection on human bronchial 2B4 cells. TMPRSS2, along with the 8 most significant proteins by absolute sum of NES, is labeled. Full results are available in Appendix A.

**Figure 4 jcm-09-00982-f004:**
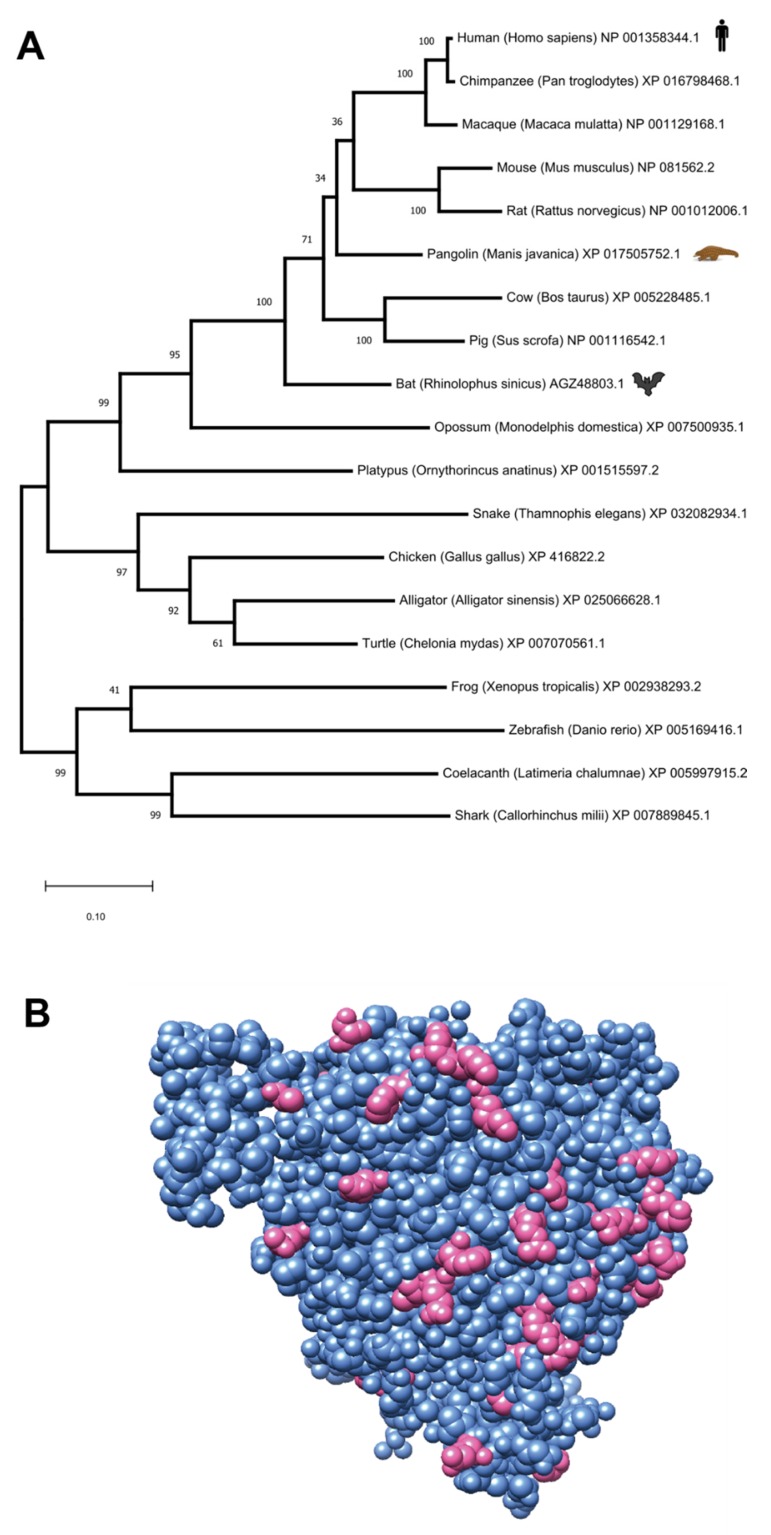
Cross-species analysis of the Angiotensin-converting enzyme 2 (ACE2) protein. (**A**) Maximum-likelihood evolutionary tree of ACE2 orthologs in selected vertebrates (numbers in the branch points indicate the % bootstrap supporting the branch structure). The tree is drawn to scale, with branch lengths measured in the number of substitutions per site. (**B**) A visualization of a human ACE2 crystal structure (resolution: 2.2 Å, PDB:1R42). The residues that are conserved across human, pangolin and bat are depicted in cornflower blue, and the residues that are conserved in human and pangolin but differ in bat (Appendix A) are depicted in opaque pink.

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
