# Peer review of "Master Regulator Analysis of the SARS-CoV-2/Human Interactome"

_jcm, 2020, doi:10.3390/jcm9040982_

Round 1

Reviewer 1 Report

The paper by Guzzi et al. describes a bioinformatics approach to identify cellular genes, which are possible targets for the development of new therapeutics against the newly emerged SARS-CoV-2. The authors combined data of an SARS-CoV-2/human interactome with transcriptome data of MERS- and SARS-CoV infected lung cells. The authors identified eight proteins, which are most affected in terms of up- or downregulation both after MERS- and SARS-CoV infection and performed a master regulator analysis to shed light on changes in expression of their interaction partners in functional proximal networks. Additionally the authors performed a phylogenetic analysis of beta coronaviruses including the newly emerged SARS-CoV-2 to clarify its relationship to the other beta coronaviruses. Finally the authors provide a cross-species analysis of ACE2 the receptor of SARS-CoV and SARS-CoV-2. They found a higher similarity of human ACE2 to pangolin ACE2 than to bat ACE2 and hypothesize that pangolins are a possible origin of SARS-CoV-2.

Although I like the idea of combining different datasets to pinpoint possible cellular targets concerning SARS-CoV-2 infection there are several shortcomings, which prevent the publication of the paper in its present form.
First of all, in my opinion the underlying concept to look for up- or downregulated genes is not the most promising one. Hoffmann et al. (doi: 10.1016/j.cell.2020.02.052) convincingly demonstrated that inhibition of TMPRSS2 with camostat mesylate inhibits SARS-CoV-2 replication. According to the authors’ results TMPRSS2 is moderately downregulated and not among their top 8 hits. On the other hand, the authors suggest MCL1, a positive regulator of apoptosis, which is highly upregulated after SARS- and MERS-CoV infection as a possible target for clinical intervention. It has been shown however, that apoptosis does not affect the viral replication kinetics of SARS-CoV (doi: 10.1007/s00705-005-0632-8).
Secondly, beta coronavirus phylogeny trees including SARS-CoV-2 leading to very similar results have been published elsewhere already (e.g. doi: 10.1016/j.cub.2020.03.022 and doi: 10.1016/j.cell.2020.02.052).
Finally, the authors conclude from their ACE2 cross species analysis that pangolins are the possible origin of SARS-CoV-2 due to the high similarity of human and pangolin ACE2. I think this conclusion is not justified since it completely misses the concept of transmission of SARS-CoV-2 via an intermediate mammalian host (e.g. bats→pangolins→humans).   

Minor points:  

p4, l154: “coded by a surprisingly compact genome of only 30kb” This sentence is wrong since the RNA genome of coronaviruses is surprisingly big. RNA genomes of other viruses are much smaller and/or segmented since the fidelity of viral RNA dependent RNA polymerases is too low to replicate larger genomes with sufficient accuracy. Coronaviruses are the exception, they encode a complex RNA dependent replication and transcription machinery which is capable to replicate a 30 kb RNA genome without incorporating too much errors.

p5, l220-21: Several groups have shown that ACE2 is indeed the SARS-CoV-2 receptor (e.g. doi.org/10.1016/j.cell.2020.02.058 and doi: 10.1016/j.cell.2020.02.052) therefore lines 220-221 on page 5 have to be changed accordingly.

p6, l274: Corticosteroids are not antiviral drugs. They can counteract virus-induced inflammation   but they do not act antiviral.

In summary, I suggest that the authors should compare their results with published intervention strategies against SARS-CoV-2, omit the beta coronavirus phylogeny and extend the ACE2 cross species analysis.

Reviewer 2 Report

In this paper, the authors studied SARS-CoV-2/host receptor recognition by performing a Master Regulator Analysis (MRA) on MERS and SARS datasets, and found that the interactions mainly affected apoptotic and mitochondrial mechanisms, and a downregulation of the ACE2 protein receptor in SARS-CoV-2 infection.  This is an interesting study with technical innovations.  However, drawing conclusions from this type of bioinformatic analyses requires added cautions, with experimental confirmation.

Comment and questions:

Line 269:  “One of the SARS-CoV-2-interacting protein networks most 268 down-regulated by beta-coronaviruses in our analysis was ACE2, and in fact the administration of 269 recombinant ACE2 to compensate this protein loss has been shown to be effective in treating acute 270 respiratory distress syndrome, which is often caused by severe pulmonary infections” ; Can authors elaborate the possibility that recombinant ACE2 binds viral particles (receptor) thus preventing the binding of virus to target cells?

Regarding the affected apoptotic and mitochondrial mechanisms, some simple “wet” lab measurement will make the case much stronger.

In result 3.2: presented data does not demonstrate a down-regulation of ACE2 by SARS-CoV-2, other than sequence similarities among these coronaviruses (Fig 4) and cross-species analysis of ACE2 proteins (Fig 5).  I am not sure how to come to the conclusion.

Line 152, from [lack of objective] [26]

Round 2

Reviewer 1 Report

In the revised version of their paper "Master Regulator Analysis of the SARS-CoV-2/Human interactome" by Guzzi et al. the authors addressed the following issues:

  • they now compare the results of their bioinformatics analysis with experimental results;
  • they removed the phylogenetic tree which was published in similar form by other groups from the main text;
  • they extended the ACE2 cross species analysis;

Since all concerns were properly addressed, I recommend the revised version for publication after correction of two minor issues in the revised manuscript:

p2l47: "deaths" instead of "death2

p5l194-195:  "An increase in mitochondria-mediated apoptotic response upon SARS-CoV infection has been reported before , although its inhibition did not seem to affect viral replication kinetics [51]." instead of "An increase in mitochondria-mediated apoptotic response has been reported before upon SARS-CoV infection, although its prevention did not seem to affect viral replication kinetics [51]."
